# Built environment impact on the per capita cycling frequency of family——Based on two-level hierarchical linear model

**Xiaonan Zhang**[1☯]*, **Jianjun Wang**[1☯], **Jianfeng Xue**[2‡], **Xueqin Long**[1‡], **Weijia Li**[1‡], **Xiaojuan Lu**[1‡], **Sai Wang**[1‡]

1 College of Transportation Engineering, Chang'an University, Xi'an, Shaanxi, China, 2 Xianyang Planning and Design Institute, Xian Yang, Shaanxi, China

☯ These authors contributed equally to this work.
‡ JX, XL, WL, XL and SW also contributed equally to this work.
* 2018021070@chd.edu.cn

**Data Availability Statement:** All relevant data are within the paper and its Supporting information files.

**Funding:** The author(s) received no specific funding for this work.

## Abstract

At present, there is less attention paid to the relationship between the frequency of travel and built environment, especially in households. In this paper, some of the determining factors in the frequency of daily cycling per household were explored based on the data from 2018 Daily Trip Survey in Xianyang, China. Then a two-level linear model was construct to identify the determining factors in the frequency of per capita daily cycling of household. According to the research results, 22.8% of the differences in the per capita cycling frequency of household are due to the differences between communities. In terms of community factors, the densities of road networks and educational facilities delivered a significantly positive impact on the per capita daily cycling frequency of family; on the contrary, the densities of medical facilities, intersections and POI delivered a significantly negative impact. Per capita cycling frequency varies considerably between households. For instance, the number of bicycles owned and the number of school-age children have a significantly positive impact on the per capita daily cycling frequency of family. However, car ownership, household income and occupation composition impose a significantly negative impact. The findings of this study would benefit the transportation engineers and planners who are keen to boost the use of active means of transportation for residents.

## 1. Introduction

As represents a comprehensive index that measures the travel demand of residents, travel ability and the level of urban traffic service, travel intensity can be assessed by a series of indicators, among which the number of trips per capita is most commonly used [1]. The number of trips mainly reflects the travel ability and demand of residents, while the average travel frequency per capita multiplied by the number of urban population is defined as the total trips, which is regarded as a basic measure of the minimum bearing capacity of urban transportation system.

**Competing interests:** The authors have declared that no competing interests exist.

Urban residents' daily trips per capita is related to their social- economic development levels, population structures, city sizes, travel modes, travel time and built environments. In addition, even the same factors may have different effects on built environments, such as household income, car (bicycle) ownership, family members, family structures, land value and population density. The first four factors are often used to explore traffic trips of family members, while land value and population density are generally used to identify the trips of traffic zones. [2].

Trip generation forecast is the first stage of the four-step traffic demand forecast method. There are many approaches for trip generation forecast, of which the most commonly used ones are category generation model, growth rate method, regression generation model, etc. [3, 4]. Miller J S et al. used four different methods to calculate the differences in trip generation rates among traffic zones in nine suburban areas, and revealed that the results acquired from the direct survey, the household survey and the trip generation rate published by the American Society of Transportation Engineers would yield quite similar results, but the traffic generation rate estimated by the regional traffic demand prediction model is significantly different from the above three methods, proving that the regional applicability of the trip generation rate should be considered [5]. Zou et al. conducted a regression model to establish the relationship between population, per capita gross domestic product and travel frequency from the overall perspective of cities [6]. On this basis, the number of trips per capita in big cities and various socioeconomic indicators were analyzed by clustering to identify four relevant indicators, including urban land area, total retail sales of consumer goods, total consumer price index and per capita disposable personal income [7]. Through a regression analysis conducted in small cities, Gao et al. proposed that the influence of travel frequency indicators can be reflected in population, area, disposable personal income and per capita GDP [8]. Yuan et al. proposed a traffic generation prediction model based on the geographical advantage of the new urban district. Furthermore, Xi'an high tech Zone was exemplified to demonstrate that the method is effective in improving the accuracy of forecasting for the travel demand [9]. However, the above model often relies on the aggregate prediction method to forecast trip generation and attraction in each traffic zone. Therefore it is difficult to reflect the impact of the socioeconomic attributes of individuals and families on the travel frequency [10].

To explore travel frequencies at individual and family levels, many scholars construct traditional regression models with individual factors, family factors and built environment factors as independent variables. Stacey has studied influence factors of travel chains [11]. Parviza et al. have explored how to use logistic models to analyze complex travel chains in Hawaii [12]. Abdelghany has established two traffic assignment models based on travel chains [13]. On the basis of the above researches, al-Jammal and Golob have constructed travel chains from the perspective of individuals and families, respectively, while simulating residents' daily travels [14, 15]. After studying the relationship between the elderly's travel frequency and the built environment, Hanibuchi et al. revealed that population densities, parks and other facilities had significantly positive impact on this frequency [16]. Agyemang-Duah K et al. found that families' geographical locations affect their weekday travels and shopping travels in Toronto, Canada, and then selected five traffic zones to establish an orderly response model, finally showing that after controlling the variables of family socioeconomic attributes, families' geographical locations affect workday travel intensities and shopping travels, with the travel intensity of suburban families slightly higher than that of downtown ones [17]. Fu et al. adopted the nonaggregate model to analyze the expected value of individual trips from the perspective of an individual [18]. Based on the trip chain theory, He et al. classified travelers into three categories to construct not only the logistic regression model for student trip, but also the Poisson distribution model for elderly trips and NL model of trip choice for employment respectively.

Moreover, the model was simulated to get trips of different categories of travelers through combination with the macro social and economic data. [19]. The default assumption in above models is that the micro-individual space is irrelevant, which ignore that each individual is also a member of a family or a community group and his/her travel behavior is also affected by the structural characteristics of his/her family and community at different levels. Therefore, it is impossible to accurately reveal the intrinsic association between built environments and travel behaviors.

In addition, with the popularization of big data, machine learning algorithms are also widely used in the prediction of travel volume, and the neural network model is a common method. Chen et al. introduced the Long Short-Term neural network to fulfill the traffic flow prediction, the results show that the traffic flow prediction accuracy of the hybrid neural network model combined with denoising method is better than that of the non-hybrid neural network model [20]. then, the data denoising scheme is combined with the deep learning model, and the EMD, EEMD and WL models of different bases are introduced to identify and correct the noise traffic flow data samples, experimental results show that the hybrid traffic flow prediction scheme is better than the prediction process [21]. However, such methods have problems such as large amount of sample data, long training time and difficult to explain.

To sum up, most of the previous studies take the traffic zone or individual as the research unit and lack of attention to family as the research unit. Furthermore, motorization-oriented generative prediction modeling has focused attention on cars, and the scale of these models is not the same as that of cycling, besides, the behaviors of groups from the same spatial are similar but there are differences between groups in different spatial regions. The existing methods ignore the intra-group homogeneity and inter-group heterogeneity of such nested data, which can lead to biased conclusions. In addition, previous studies paid more attention to the built environment and population density, traffic facility density, POI diversity and other conventional indicators. The safety of riding facilities and the influence of different types of facility density on riding facilities need to be further studied.

In order to overcome the limitations of most of the previous studies, this study construct a two-level linear model based on data from 2018 Daily Trip Survey in Xianyang, China and select the family attributes and built environment as dependent variables, to explore the influence of spatial heterogeneity on travel frequency. This study not only provides a comprehensive overview of factors that influence per capita cycling frequency of family, but also quantifies the spatial heterogeneity of the cycling frequency and reveal the role of communities' built environments in families' cycling frequency. The results of the study can provide technical support for urban planning, policy designing, boosting green urban travels, reducing transportation energy consumption and carbon emissions, and make cities healthier and more sustainable places to live and work.

This paper is organized as follows. The next section illustrates the data sources and the influencing factors, as well as indicators to measure built environment. Section 3 introduce the research methodology and the zero model, random intercept model and full model are constructed. Section 4 analyzes the model results, and the effect of different level factors are further explicated. The results are discussed in the last section.

## 2. Material and research methodology

### 2.1 Study area and data sources

As part of Shaanxi Province, Xianyang city is located in northwest China and adjacent to the provincial capital Xi 'an in the east. Covering a total of 10196 square kilometers, it has a permanent population of 395,9842 and an annual GDP of CNY 220.481 billion. Taken as research

object for this study, the relatively well-developed city center of Xianyang covers an area of 60 square kilometers, including 760000 people. The travel data was obtained through a two-stage stratified sampling survey. In the first one, the whole sampling method was applied to select the communities to be surveyed. In the second and last one, the survey households were randomly selected from each community. Besides, all-day travel survey of family members aged over 6 years was conducted in the form of home visits. It should be noted that, in the first page of the questionnaire, we introduced our research purpose and secured their written informed consent to participate in this academic research. Non-adult participants were allowed to answer this questionnaire only after getting their parent or guardian's written informed consent. The Chang'an University Ethics Committee approved the protocol and informed consent forms for this study.

The household survey was conducted during the period ranging from May 29 to May 31, 2018, with a total of 5,080 households and 13,607 respondents investigated and a success rate of 100% reached. The dependent variable in this study is the per capita cycling frequency of family. 108 communities and 2,934 families were found meeting the requirements of this research, and the minimum sample of families in each community was not less than 10. According to Goldstein, the sample size should be no less than 10 for each group and no less than 100 for level 2 to reduce sampling errors and balance the sample size between groups [22]. Given that there were 108 communities in level 2, each with a samples of 10 or more families, the sample size meets the requirements of subsequent model analyses. The community boundaries as demonstrated for the research units are shown in S1 Fig.

## 2.2 Variable selection and description

**2.2.1 Family socio-economic attributes.** According to previous studies, family structure, family income, household car and bicycle ownership are all associated with travel behavior [23–26]. The household socioeconomic attributes include total household population(PEO), household occupational composition(PRO), monthly household income(INCOM), the number of school-age children(CHIL), the number of elderly(OLD), ownership of vehicles and other attributes may affect travel behavior. Variables description is shown in Table 1.

As can be seen from Table 1, the majority of the families have 2–3 members, mainly with a double-worker composition, followed by single-worker. And jobless families take the lowest proportion. The largest part of the families report a monthly income of CNY3001–6000, representing 78%, followed by below CNY3000, representing 17.5%, high income families (with monthly income of above CNY10000) account for 4.5%. For the ownership of transportation, 55% of households have at least one car, and 58.2% of households have at least one bicycle, families with two school-age children and one school-age child had almost the equal proportion.

**2.2.2 Built environment.** In 1997, Cervero and Kockelman proposed a "3D" (density, diversity and design) model for the measurement of built environment [27, 28]. Then, Cervero et al. introduced accessibility and distance into 3D elements and transformed them into a "5D" model for studying Bogota built environment and its impact on walking and cycling behavior. Other scholars used "5D"model to study the correlation between travel characteristics and built environment in subsequent studies [29, 30]. Combination with the availability of data, this study divides the built environment variables into three categories: community population density, road traffic infrastructure and other daily life service facilities, including 12 built environment indicators.

1. Community population density
Population density is regarded as an important influencing factor towards travel intensity.

**Table 1. Descriptive statistics for family socio-economic attributes (N = 2934).**

| Variable | Question options | n | Percentage (%) | Variable | Question options | n | Percentage (%) |
|---|---|---|---|---|---|---|---|
| **PEO** | 1 | 110 | 3.7 | **OLD** | 0 | 2448 | 83.4 |
| | 2 | 1082 | 36.9 | | 1 | 186 | 6.3 |
| | 3 | 1443 | 49.2 | | 2 | 300 | 10.3 |
| | 4 | 234 | 8.0 | **CHIL** | 0 | 1519 | 51.8 |
| | 5 | 58 | 2.0 | | 1 | 633 | 21.6 |
| | 6 | 7 | 0.2 | | 2 | 649 | 22.1 |
| **BIKE** | 0 | 1217 | 41.5 | | 3 | 133 | 4.5 |
| | 1 | 1313 | 44.8 | **INCOM** | <CNY3000 | 513 | 17.5 |
| | 2 | 379 | 12.9 | | CNY3001-10000- | 2290 | 78.0 |
| | 3 | 25 | 0.8 | | >CNY10000 | 131 | 4.5 |
| **CAR** | 0 | 1320 | 45.0 | **PRO** | double workers | 1774 | 60.5 |
| | 1 | 1555 | 53.0 | | single workers | 760 | 25.9 |
| | 2 | 58 | 2.0 | | jobless | 400 | 13.6 |
| | 3 | 1 | 0.0 | | | | |

Note: CHIL refers to primary, middle and high school students aged between 6 and 18.

An empirical study on the influence of urban built environment on commuting mode choice in Shanghai shows that the population density is positively correlated with the probability of non-motor vehicle use [31]. According to a study in Maryland, the higher the population density is, the higher the possibility of carrying bicycle is, In place with higher industrial employment, the use of bicycles increase relatively [32]. A study on the relationship between built environment and physical activity shows that the population density, parks or green spaces are positively correlated with activity frequency regardless of the buffer zone chosen [16]. In this study, the population density was calculated by the total population in a community /community area.

2. Road transportation infrastructure

In the most relevant studies, the road infrastructure indicators mainly included road network density, the road section, intersection density, bus station density, subway station density, etc. [33, 34]. This study used the road network density, intersection density and bus station density as the indicators of road traffic infrastructure, which were calculated as follows:

Road network density = the total length of road network in a community/the community area.

Intersection density = the number of intersections in a community/the community area.

Bus station density = the total number of bus stations in the community/the community area.

3. Other daily life service facilities

Many studies take land-use density and land-use diversity as quantitative indicators of other service facilities in the built environment and obtain consistent conclusions by analyzing the relationship between them and travel behavior [35–37]. A few studies used POI as an indicator of built environment [38, 39]. This study took POI density, POI diversity, and the POI density of four main daily life facilities (including school, shopping, leisure,

and medical facilities) as the indicators of other daily life service facilities, as follows:

The POI density of daily life service facilities = the total number of POIs of the $i-$ th facility in a community/the community area.

POI density = $\Sigma$ total number of POIs of the $i-$ th facility in a community/the community area.

Based on the existing literature [40], the community POI diversity is defined as follows:

$$H_s = -\sum_i^M \sum_j^N P_{ij} \times log P_{ij} \tag{1}$$

$$P_{ij} = A_{ij}/A_k \tag{2}$$

$$\sum_i \sum_j P_{ij} = 1 \tag{3}$$

Where $P_{ij}$ represents the probability, $A_{ij}$ indicates the number of type $j$ POI facilities in the i-th community, and $A_k$ denotes the total number of POIs in the i-th community. In the formula, $H_s$ refers to the spatial information entropy. It can be find out that when $H_s \geq 0$, the level of spatial information entropy can be used to indicate the mixing degree of various facilities in the community. That is to say, higher entropy indicates more types of facilities.

The following figure (S2–S10 Figs) showed the attribute information about population density(S2), POI diversity(S3), POI density(S4), road network density(S5), intersection density (S6), bus station density(S7), the POI density of education, shopping, leisure, and the medical services(S8), the proportion of motor and non-motor vehicle separate railing (S9) and the proportion of non-motor lane parking ratio(S10) in each community.

As can be seen from S2, the population density gradually decreases from the center to the periphery, and the population density in the areas with dense commercial service facilities is higher than that in the new urban areas. It can be seen from S3 and S4 that both the POI diversity and POI density show a typical pattern of contiguous distribution and spread along the line. Contiguous distribution mainly includes the old street area, the business circles on Maotiao Road, the business circles of 505 Square an others, while the spread line mainly includes Renmin Road, Yuquan Road, Weiyang Road, etc. As for the new urban areas, both the diversity and density of POI are low.

Seen from S5, S6, S7 and S8, the densities of road network, intersection, bus station and public service facility around the old urban areas and commercial core areas are higher than those of the urban fringe areas and newly built areas. S8 and S9 show that the traffic volume of the commercial core areas is large and most of the roads are equipped with motor and non-motor vehicle separate railings. The roads of the new areas in the west and south were built under higher construction standards with cross section of three or four boards roads in the main and secondary roads, therefore, the motor and non-motor vehicle separate railings account for a relatively high proportion. From S10, it can be seen that the proportion of non-motor lane parking in the western part of the built-up area is more serious than that in the eastern part. For one thing, there are relatively few traffic monitoring facilities in the west and the law enforcement is weaker than that in the east. For another, there are fewer public parking lots in the west areas compared with the east areas.

**2.2.3 Per capita cycling frequency of family(PCF).** The existing studies usually took different periods of travel time (week, day, peak hour or hour), arrival/departure rates and travel frequency as dependent variables [41, 42]. However, this research mainly studies the influence of community built environment on family cycling frequency. The relationship between

cycling frequency and built environment is more significant than that between cycling time and built environment [43]. Therefore, take the per capita cycling frequency of family(PCF) as dependent variable, which was calculated by dividing the total daily cycling frequency of family members by the number of family members over 6 years old.

## 2.3 Methodology

In order to study the effect of family social-economic characteristics and built environment on the per capita cycling frequency of family, a two-level hierarchical linear model (HLM) was applied. Due to the nested data structure of family travel frequency within community, HLM was considered as the most appropriate method that can be used to analyze this data. All analyses were conducted using HLM7 software. The Level-1 variables included the total household population, household income, the number of children, the number of the old and the occupation of family member. The Level-2 variables included the density of community population, the density of road network, intersection density, the density of bus station, POI diversity, POI density and the density of education, medical, shopping and leisure facilities, all of which were closely related to the daily life of residents. The community ID corresponding to each family represented the unique connection point between the variables of level-1 and level-2.

Modeling a multi-layer model is a step-by-step process. Combining the methods recommended by Hox (1994) and Singer (1998), the construction process consists of the following three mode is: null mode, random intercept model and full mode. The null model can be used to assess within-group or between-group homogeneity and only when there is significant intra-group correlations between them, then a multi-level linear model can be established. The random intercept model includes random effect regression models as well as regression models that use means as outcome measures. This model can explore the influence of household level variables and regional level variables on the per capita cycling frequency of family. The full model is used to explore whether the level-2 variables deliver a significant impact on the intercept and coefficient of the level-1 model.

**2.3.1 Null model.**   Null model or intercept-only model, which is identical to a one-way random effect analysis of variance, the general form of null model is as follows.

$$Level - 1 : \ Y_{ij} = \beta_{0j} + r_{ij} \tag{4}$$

$$Level - 2 : \ \beta_{0j} = \gamma_{00} + \mu_{0j} \tag{5}$$

While, In level-1 $Y_{0j}$ indicates the per capita cycling frequency of family(PCF), $\beta_{0j}$ represent mean outcome of group $j$, $r_{ij}$ represent the random individual variable around this mean. $\gamma_{00}$ denotes the overall intercept representing the grand mean of $Y_{ij}$, $\mu_{0j}$ captures the variable between group means.

**2.3.2 Random intercept model.**   Random intercept models include random effects regression models and regression models with the mean as the outcome variable. Thereinto, the random intercept regression model is used to study the level-1 variables, that is, the effect of family level factors on cycling frequency; the random effects regression model is used to study the level-2 variables, that is, the effect of community level factors on cycling frequency.

• Random effects regression model

The random effects regression model only involved the family level variables and was used to test the correlation between the level-1 variables and the outcome variables. The random

effects regression model was established as follows:

$$Level-1: \ Y_{ij} = \beta_{0j} + \beta_{1j}PEO + \beta_{2j}CAR + \beta_{3j}BIKE + \beta_{4j}CHIL + \beta_{5j}OLD +$$
$$\beta_{6j}INCOM1 + \beta_{7j}INCOM3 + \beta_{8j}PRO2 + \beta_{9j}PRO3 + r_{ij} \tag{6}$$

$$Level-2: \ \beta_{0j} = \gamma_{00} + \mu_{0j} \tag{7}$$

$$\beta_{1j} = \gamma_{10} \tag{8}$$

$$\beta_{2j} = \gamma_{20} \tag{9}$$

$$\beta_{3j} = \gamma_{30} \tag{10}$$

$$\beta_{4j} = \gamma_{40} \tag{11}$$

$$\beta_{5j} = \gamma_{50} \tag{12}$$

$$\beta_{6j} = \gamma_{60} \tag{13}$$

$$\beta_{7j} = \gamma_{70} \tag{14}$$

$$\beta_{8j} = \gamma_{80} \tag{15}$$

$$\beta_{9j} = \gamma_{90} \tag{16}$$

While, $\beta_{0j}$ in level-1 represents the average result of the $j$-th unit, $r_{ij}$ denotes the error of each item in level-1, $\beta_{1j}, \beta_{2j}, \beta_{3j}, \beta_{4j}, \beta_{5j}, \beta_{6j}, \beta_{7j}, \beta_{8j}$ and $\beta_{9j}$, respectively indicate the total household impact coefficients of population, family car ownership, family bicycle ownership, number of school-age children in the family, number of elderly in the family, low-income families, high-income families, single-employee families, and workless families. $\gamma_{00}$ in level-2 refers to the overall mean of the results, and $\mu_{0j}$ refers to the random effect associated with the $j$-th unit.

• Regression model with the mean as the result

In order to further analyzed the impact of community level factors on the per capita cycling frequency of family, this paper used the average of the level-2 variables as the estimation result to establish a regression model, which was as follows:

$$Level-1: \ Y_{ij} = \beta_{0j} + r_{ij} \tag{17}$$

$$Level-2 \ \beta_{0j} = \gamma_{00} + \gamma_{01}PCPEO + \gamma_{02}POID + \gamma_{03}PPOI + \gamma_{04}PNET + \gamma_{05}PINT +$$
$$\gamma_{06}PBUS + \gamma_{07}PEDU + \gamma_{08}PSHO + \gamma_{09}PLEI + \gamma_{10}PMED + \gamma_{11}PISO + \gamma_{12}PPOK + \mu_{0j} \tag{18}$$

While, $\beta_{0j}$ in level-1 represents the average result of the $j$-th unit, and $r_{ij}$ denotes the error of each item in level-1.

$\gamma_{00}$ in level-2 refers to the overall mean value of the result, $\gamma_{00}, \gamma_{01}, \gamma_{02}, \gamma_{03}, \gamma_{04}, \gamma_{05}, \gamma_{06}, \gamma_{07}, \gamma_{08}, \gamma_{09}, \gamma_{10}, \gamma_{11}$ and $\gamma_{12}$ respectively indicate community population density, community POI diversity, and community POI density, road network density, intersection density, bus stop

density, POI density of education, shopping, leisure and medical, proportion of non-isolation bar, and proportion of non-motor vehicle parking. $\mu_{0j}$ refers to the conditional variance of $\beta_{0j}$ after the control variable.

**2.3.3 Full model.** The full model was used to study the average intercept of the level-1 model and the variation of coefficients at the level-2. The level-1 and level-2 variables of random intercept model are brought into the full model for modeling. The specific model was as follows:

$$Level - 1 : \ Y_{ij} = \beta_{0j} + \beta_{1j}CAR + \beta_{2j}BIKE + \beta_{3j}CHIL + \beta_{4j}INCOM1 + \beta_{5j}INCOM3 \\ + \beta_{6j}PRO2 + \beta_{7j}PRO3 + r_{ij} \tag{19}$$

$$Level - 2 : \ \beta_{0j} = \gamma_{00} + \gamma_{01}POID + \gamma_{02}PNET + \gamma_{03}PINT + \gamma_{04}PEDU + \gamma_{05}PMED + \mu_{0j} \tag{20}$$

$$\beta_{0j} = \gamma_{00} + \mu_{0j} \tag{21}$$

$$\beta_{1j} = \gamma_{10} \tag{22}$$

$$\beta_{2j} = \gamma_{20} + \gamma_{21}PNET \tag{23}$$

$$\beta_{3j} = \gamma_{30} + \gamma_{31}PNET \tag{24}$$

$$\beta_{4j} = \gamma_{40} + \gamma_{41}POID + \gamma_{42}PNET \tag{25}$$

$$\beta_{5j} = \gamma_{50} + \gamma_{51}POID + \gamma_{52}PNET \tag{26}$$

$$\beta_{6j} = \gamma_{60} + \gamma_{61}POID + \gamma_{62}PEDU \tag{27}$$

$$\beta_{7j} = \gamma_{70} + \gamma_{71}POID + \gamma_{72}PEDU \tag{28}$$

Where, $\beta_{0j}$ in level-1 represents the average result of the j-th unit, $\beta_{1j}, \beta_{2j}, \beta_{3j}, \beta_{4j}, \beta_{5j}, \beta_{6j}$ and $\beta_{7j}$ respectively indicate the total family population, family car ownership, family bicycle ownership, family school-age children, number of elderly in the family, low-income households, high-income households, single-employee households, and workless households. $r_{ij}$ denotes the error of each item in level-1. $\gamma_{00}$ in level-2 refers the average intercept of all units in level-2. $\gamma_{00}, \gamma_{01}, \gamma_{02}, \gamma_{03}, \gamma_{04}, \gamma_{05}$ respectively indicate the community POI diversity, road network density, intersection density, POI density of education and medical on the per capita cycling frequency of household, $\gamma_{00}, \gamma_{10}, \gamma_{20}, \gamma_{30}, \gamma_{40}, \gamma_{50}, \gamma_{60}, \gamma_{70}$ refer the average regression slopes of all units in level-2, $\gamma_{20}$ and $\gamma_{31}$ indicate the influence coefficients of road network density on the bicycle ownership and school-age children riding in level-1, $\gamma_{41}, \gamma_{51}, \gamma_{61}$ and $\gamma_{71}$ indicate the influence coefficient of community POI diversity on the cycling frequency of high and low-income families in level-1, and $\gamma_{42}$ and $\gamma_{52}$ denote the impact coefficient of road network density on the cycling frequency of high- and low-income families in level-1. $\gamma_{62}$ and $\gamma_{72}$ denote the influence coefficients of educational facilities on the cycling of single-employees and unemployed families on level-1.

**Table 2. Parameter estimation of null model on the per capita cycling frequency of family.**

| Random Effect | Standard Deviation | Variance Component | Df | Chi-square | P-value |
|---|---|---|---|---|---|
| U0 | 0.577 | 0.33 | 107 | 947.11 | 0.000 |
| R | 1.057 | 1.12 | | | |

Note: ICC = 0.228.

## 3. Results

### 3.1 Null model estimation

To begin with, an unconditional model was tested to determine whether there were significant variation existing at each level of the hierarchy. The parameter estimation were shown in Table 2, from which can be seen that there was a significant differences between groups ($\hat{\sigma}_{\mu 0}^2 = 0.33, p = 0.000$). That was to say, there were significant inexplicable differences among community.

According to Cohen, an ICC is lower than 0.059 which indicates a weak intra-group correlation, the range from 0.059 to 0.138 suggests a moderate correlation, and an ICC higher than 0.138 which indicates a strong correlation. In general, as long as ICC(1) exceeds 0.059, it is necessary to consider inter-group effects in statistical modeling processing [44]. The intra-group correlation coefficient (ICC) of the per capita cycling frequency of family was 0.228, which means that about 22.8% of the total variation in the outcome measure was attributed to the variations between communities, constituting a significant intra-group correlation. In this case, it was not suitable to adopt the general regression model and a multilevel model was required for data analysis.

### 3.2 Random intercept model estimation

**3.2.1 Random effects regression model estimation.** Based on null model, a random coefficient regression model was introduced, under which only the level-1 had predictor variables, while level-2 had none. The level-1 predictor variables included the total family population, the number of owned cars, the number of owned bicycles, the number of school-age children in a family, the number of elders in a family, the monthly household income, and the family occupational composition. The parameter estimation of the random coefficient regression model was shown in Table 3.

The results demonstrated that the variables included the total household population and the number of elderly in level-1 had no significant impact on the per capita cycling frequency of family, while other variables included car ownership, bicycle ownership, number of school-age children in a family, family income and family occupation composition had statistically significant impact on the per capita cycling frequency of family.

It can be seen from Table 3 that in the level-1 predictor variables, car ownership had a significant negative impact on the per capita cycling frequency of family(sig. = 0.000<0.001), That was to say, the car ownership increases by one unit, the per capita cycling frequency of family was reduced by 0.359 units. while, bicycle ownership had a significant positive impact on the per capita cycling frequency of family (sig. = 0.000<0.001), suggesting that the number of family bicycles increased by one unit, the per capita cycling frequency of family increased by 0.367 unit; the number of school-age children in the family had a significant impact on the frequency of family cycling (sig. = 0.006<0.01, regression coefficient is 0.080). There was a significant difference in the per capita cycling frequency of family between low-income, high-income families

Table 3. The parameter estimation of the random coefficient regression model.

| Fixed effect | Coefficient | Standard error | T-ratio | d.f. | P-value |
|---|---|---|---|---|---|
| INTRCPT1 | 1.431 | 0.130 | 10.999 | 107 | 0.000 |
| PEO | 0.056 | 0.040 | 1.423 | 2924 | 0.152 |
| CAR | -0.359 | 0.048 | -7.467 | 2924 | 0.000 |
| BIKE | 0.367 | 0.040 | 9.151 | 2924 | 0.000 |
| CHIL | 0.080 | 0.029 | 2.790 | 2924 | 0.006 |
| OLD | 0.011 | 0.047 | 0.226 | 2924 | 0.821 |
| INCOM1 | 0.239 | 0.071 | 3.378 | 2924 | 0.001 |
| INCOM3 | -0.183 | 0.072 | -2.557 | 2924 | 0.011 |
| PRO2 | 0.150 | 0.051 | 2.939 | 2924 | 0.004 |
| PRO3 | 0.301 | 0.101 | 2.963 | 2924 | 0.004 |

Note: The parameter estimates was based on robust standard error

and middle-income families. (Sig. = 0.001<0.01 and sig. = 0.011<0.05), among them, the per capita cycling frequency of low-income families was 0.239 units higher than that of middle-income families and the per capita cycling frequency of high-income families was 0.183 units lower; single-employee families were significant differences in the per capita cycling frequency between workless families and dual-employee families (sig. = 0.004<0.01 and sig. = 0.004<0.01), and the per capita cycling frequency of single-employee families and workless families were 0.150 and 0.301 units higher than that of dual-employee families, respectively.

**3.2.2 Regression model with the mean as the result estimation.** On the basis of the zero model estimation, the level-2 had predictor variables and the level-1 had none. The level-2 predictive variables included community population density, community POI diversity, community POI density, road network density, intersection density, bus station density, POI density of education, shopping, leisure, medical, the proportion of non-isolation bar and the proportion of non-motor lane parking ratio. The parameter estimation of the regression model based on the mean value was shown in Table 4.

The results showed that in the level-2 predictor variables included the community population density, POI density, bus station density, shopping facilities, leisure facilities, the proportion of non-isolation bar and the proportion of non-motor lane parking ratio had no significant impact on the per capita cycling frequency of family, while other level-2 predictive variables included POI diversity, road network density, intersection density, educational facilities and medical facility density had reached Statistically significant level.

It can be seen from Table 4 that in the level-2 predictor variables, POI diversity, intersection density, and medical facilities had a significant negative impact on the per capita cycling frequency of family(sig. = 0.003<0.01, sig. = 0.034<0.05 And sig. = 0.007<0.01), that was, for each additional unit in POI diversity, intersection density, and medical facility density, the per capita cycling frequency of household would be decrease by 0.823, 0.003, and 0.002 units respectively, the road network density and the density of education facilities had a significant positive impact (sig. = 0.008<0.01 and sig. = 0.000<0.001), that was, for each additional in road network density and educational facility density, the per capita cycling frequency of family would be increased by 0.003 and 0.021 unit respectively.

## 3.3 Full model estimation

According to the results of the random coefficient regression model, the variables in the level-1 that had no significant impact on the per capita cycling frequency of families were

Table 4. The parameter estimation of the regression model based on the mean value.

| Fixed Effect | Coefficient | Standard error | T-ratio | d.f. | P-value |
|---|---|---|---|---|---|
| INTRCPT1 | 1.832 | 0.055 | 33.368 | 95 | 0.000 |
| PCPEO | -0.000 | 0.000 | -0.678 | 95 | 0.499 |
| POID | -0.823 | 0.262 | -3.145 | 95 | 0.003 |
| PPOI | 0.121 | 0.081 | 1.491 | 95 | 0.139 |
| PNET | 0.003 | 0.001 | 2.735 | 95 | 0.008 |
| PINT | -0.003 | 0.001 | -2.143 | 95 | 0.034 |
| PBUS | 0.001 | 0.003 | 0.217 | 95 | 0.829 |
| PEDU | 0.021 | 0.005 | 4.315 | 95 | 0.000 |
| PSHO | 0.007 | 0.045 | 0.152 | 95 | 0.880 |
| PLEI | -0.001 | 0.004 | -0.197 | 95 | 0.844 |
| PMED | -0.002 | 0.001 | -2.761 | 95 | 0.007 |
| PPOK | 0.129 | 0.090 | 1.430 | 95 | 0.156 |
| PISO | 0.006 | 0.009 | 0.655 | 95 | 0.514 |

Note: The parameter estimates in the table are based on robust standard error

eliminated, while those with significant influences variables were retained. Therefore, the predictor variables that were adopted in the full model were: car ownership, bicycle ownership, the number of school-age children in a family, family income, and family occupational composition. Similarly, predictive variables in the level-2, including POI diversity, road network density, intersection density, education facility density and medical facility density were kept in the full model.

The parameter estimates of the full model are shown in Table 5, which suggested that the diversity of POI and road network on the community level had no significant impact on the per capita cycling frequency of family whereas the density of intersections, educational and medical facilities exerted a significant effect on the per capita cycling frequency of family. Besides, the density of educational facilities exhibited a significant positive impact on the per capita cycling frequency of family as controlling for other relevant variables(p = 0.000<0.001). That is to say, for each unit increase in the density of educational facilities, the per capita cycling frequency of family would be increased by 0.015 unit, suggesting that educational facilities can be used to partially explain the differences in the per capita cycling for families in the community. Moreover, the density of intersection and medical facility had a significant impact on the per capita cycling frequency of family(p = 0.000<0.001 and P = 0.009<0.01). Contrary to what was expected, the density of intersection and medical facility had a significant impact on the per capita cycling frequency of family. One possible cause for this is that the high density of intersections may lead to frequent cycling interruptions. And furthermore, the medical facilities were usually used in case of physical discomfort, therefore, most of the trips relay on motor vehicles.

Among the level-1 predictor variables, the household car ownership still had a significantly negative impact on the per capita cycling frequency of family (sig. = 0.000<0.001) with a regression coefficient of -0.346, indicating that the per capita cycling frequency of family would be reduced by 0.346 units with per one unit increase in household car ownership. The household bicycle ownership and the number of school-age children in a family still had a significantly positive impact on the per capita cycling frequency of family (sig. = 0.000<0.001 and sig. = 0.000< 0.001) with regression coefficients of 0.370 and 0.119, respectively. The more the bicycles and school-age children in a family had, the higher the per capita cycling frequency of

Table 5. The parameter estimates of the full model.

| Fixed Effect | Coefficient | Standard error | T-ratio | d.f. | P-value |
|---|---|---|---|---|---|
| INTRCPT | 1.547 | 0.087 | 17.712 | 102 | 0.000 |
| POID | -0.281 | 0.199 | -1.413 | 102 | 0.161 |
| PNET | -0.001 | 0.002 | 0.710 | 102 | 0.479 |
| PINT | -0.004 | 0.001 | -4.104 | 102 | 0.000 |
| PEDU | 0.015 | 0.004 | 4.100 | 102 | 0.000 |
| PMED | -0.001 | 0.001 | -2.677 | 102 | 0.009 |
| CAR | -0.346 | 0.048 | -7.246 | 2911 | 0.000 |
| BIKE | 0.370 | 0.039 | 9.548 | 2911 | 0.000 |
| BIKE*PNET | 0.001 | 0.000 | 2.621 | 2911 | 0.009 |
| CHIL | 0.119 | 0.026 | 4.479 | 2911 | 0.000 |
| CHIL*PNET | 0.002 | 0.001 | 2.127 | 2911 | 0.033 |
| INCOM1 | 0.213 | 0.068 | 3.139 | 2911 | 0.002 |
| INCOM1*POID | -0.368 | 0.370 | -0.994 | 2911 | 0.321 |
| INCOM1*PNET | -0.002 | 0.002 | -1.139 | 2911 | 0.255 |
| INCOM3 | -0.163 | 0.065 | -2.524 | 2911 | 0.012 |
| INCOM3*POID | -0.455 | 0.099 | -4.577 | 2911 | 0.000 |
| INCOM3*PNET | 0.004 | 0.003 | 1.289 | 2911 | 0.198 |
| PRO2 | 0.147 | 0.050 | 2.971 | 2911 | 0.003 |
| PRO2*POID | 0.160 | 0.169 | 0.946 | 2911 | 0.345 |
| PRO2*PEDU | 0.008 | 0.005 | 1.726 | 2911 | 0.084 |
| PRO3 | 0.309 | 0.080 | 3.878 | 2911 | 0.000 |
| PRO3*POID | 0.460 | 0.213 | 2.159 | 2911 | 0.031 |
| PRO3*PEDU | 0.008 | 0.002 | 3.680 | 2911 | 0.000 |

Note: The parameter estimates in the table are based on robust standard error

family was. The frequency of low-income and high-income families was significantly different from that of middle-income ones (sig. = 0.002<0.01 and sig. = 0.012<0.05). In addition, the per capita cycling frequency of low-income families was 0.213 units higher than that of middle-income ones, while the per capita cycling frequency of high-income families was 0.163 units lower. There was a significant difference in the per capita cycling frequency between single-employee, workless and dual-employee families (sig. = 0.003<0.01 and sig. = 0.000<0.01). The per capita cycling frequency of single-employee and workless families were 0.147 and 0.309 units higher than that of dual-employee families, respectively. To sum uo, family factors delivered significantly greater impact on a family's per capita cycling than community factors.

On cross-level interaction, the density of road network promoted cycling behavior among those families with bicycles and school-age children. Statistically, the cross-level interaction coefficients between road network density and bicycle ownership, the number of school-age children were 0.001 (sig. = 0.009<0.01) and 0.002 (sig. = 0.033<0.05), while the interaction between POI diversity and high-income households had a more significant negative impact compared with middle-income households, with the interaction coefficient reaching -0.455 (sig. = 0.000< 0.001). Besides, the interaction between POI diversity and jobless families exerted a more significant positive effect compared with dual-income families, with the interaction coefficient reaching 0.460 (sig. = 0.031<0.05). In addition, the density of educational facilities showed a significant difference in promoting cycling behavior between jobless families and dual employees, with the interaction coefficient reaching 0.008 (sig. = 0.000< 0.001).

This may result from the fact that jobless families tended to choose the school closer to their residence and the children from this family were more inclined to cycling. In comparison, the dual-income families have higher incomes and pay more attention to the quality of educational facilities, with the children from these families are usually driven to and from school by their parents.

## 4 Conclusion and discussion

Analysis based on survey data, this study employ a two-level linear model to identify the main factors affecting the per capita cycling frequency of family, which focus on the impact and interaction of this frequency at two different levels of the family and the community. The results show that 22.8% of the heterogeneity in the per capita cycling frequency of family comes from communities, which means that more differences among these families. That is to say, family factors are more important in explaining the differences in the households per capita daily cycling frequency, because each individual trip in a family are often restricted by the family social-economic attributes and the division of among family members, while the influence of communities is relatively weak. This results are consistent with the findings of international research. Surveys on Shanghai residents' cycling reveal that the frequency of using shared bicycles are mainly affected by age, education, income and family size, etc., followed by riders' satisfaction with the riding environment safety and comfort [19]. W. Wendel-Vos1's research demonstrates that a few hypothetical environments have been supported as determinants of active travels such as cycling [45]. Ding studied the impact of family structures and built environments on travel behavior of the elderly and non-elderly and the results showed that whether living with children, population density and the degree of mixed land use affect the travel characteristics of the elderly [46].

At family factor aspect, the bicycle ownership and the number of school-age children in the family have a significantly positive impact on the per capita cycling frequency of family, while the car ownership, family income and family occupation composition have a significantly negative impact on the per capita cycling frequency of family. That is to say, the relevant policies such as raising the threshold of vehicle ownership can improve the frequency of cycling.

As far as community factors are concerned, the density of educational facilities has a significantly positive impact on the per capita cycling frequency of family, suggesting that measures such as enriching the education facilities in communities would help boost cycling. Contrary to what is expected, both the POI density of medical facilities and the intersection density have significantly negative impact on the per capita cycling frequency of family. One possible explanation for this phenomenon was that the high intersection density may make frequent interruptions for cycling; another possible explanation is that the medical facilities were required in case of physical discomfort, therefore, most of trips relay on motor vehicles.

Then a cross-level interactions indicate that the density of road network promote cycling in families with bicycles and school-age children. In addition, the density of educational facilities has a more significant difference in promoting cycling among the jobless families than that of dual employees, which may be attributed to the fact that jobless families tend to choose the school closer to their residence and the children from this family are more inclined to cycling. In contrast, the dual-income families have higher incomes and pay more attention to the quality of educational facilities and the children from these families are usually driven to and from school by their parents.

To sum up, diversity of public service facilities, good road infrastructure and appropriate policies are conductive to the cycling safety and comfort for community members to promote cycling. Same specific measures are as follows: firstly, on the urban construction, reasonable

arrangement of education, medical and other facilities adjacent to home areas can ensure that daily travel distances of residents are within a proper range of cycling. Secondly, it is necessary to improve the safety of cycling environment by establishing bicycle lanes and other facilities. Finally, if the existing built environment is difficult to optimize, policies such as increasing the barrier to purchasing cars shall be considered, so as to curb the use of motor vehicles and promote active travel behavior.

This study also has some limitations, Firstly, the model can be further expanded into a three-level model by adding individual level factors in the future. Secondly, other factors such as residents' personal values, group identity, weather, as well as terrain can be considered in the future research. Finally, the sample size can be further expanded and the groups can be divided with travel purposes and age stages, which will help transportation engineers and planners to put forward feasible strategy and targeted tactics to increase the use of active means of transportation for residents.

## Supporting information

**S1 Fig. Research scope and community boundaries.**
(TIF)

**S2 Fig. Population density.**
(TIF)

**S3 Fig. POI diversity.**
(TIF)

**S4 Fig. POI density.**
(TIF)

**S5 Fig. Road network density.**
(TIF)

**S6 Fig. Intersection density.**
(TIF)

**S7 Fig. Bus station density.**
(TIF)

**S8 Fig. The POI of shopping, leisure, education and medical.**
(TIF)

**S9 Fig. The proportion of non-isolation bar.**
(TIF)

**S10 Fig. The proportion of non-motor lane parking ratio.**
(TIF)

**S1 Data.**
(XLSX)

## Acknowledgments

The authors would like to thank Xianyang Planning and Design Institute and each member of the 905 research group for data collection and data processing. The authors also thank the respondents for providing data and information that were essential for this work.

## Author Contributions

**Conceptualization:** Xiaonan Zhang.

**Investigation:** Jianfeng Xue, Xueqin Long, Weijia Li, Xiaojuan Lu, Sai Wang.

**Methodology:** Xiaonan Zhang.

**Software:** Xiaonan Zhang.

**Supervision:** Jianjun Wang.

**Visualization:** Xueqin Long.

**Writing – original draft:** Xiaonan Zhang.

**Writing – review & editing:** Xiaonan Zhang.

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
