## [Decision Letter · Decision Letter 0]

21 Feb 2022

PONE-D-21-40172Built Environment Impact on the Per Capita Cycling Frequency of FamilyPLOS ONE

Dear Dr. Zhang,

Thank you for submitting your manuscript to PLOS ONE. After careful consideration, we feel that it has merit but does not fully meet PLOS ONE’s publication criteria as it currently stands. Therefore, we invite you to submit a revised version of the manuscript that addresses the points raised during the review process.

Please revise the manuscript carefully to address all the reviewers' comments and concerns.

We look forward to receiving your revised manuscript.

Kind regards,

Quan Yuan, Ph.D.

Academic Editor

PLOS ONE

Journal Requirements:

Reviewers' comments:

Reviewer's Responses to Questions

**Comments to the Author**

1. Is the manuscript technically sound, and do the data support the conclusions?

Reviewer #1: Partly

Reviewer #2: Partly

2. Has the statistical analysis been performed appropriately and rigorously? 

Reviewer #1: Yes

Reviewer #2: Yes

3. Have the authors made all data underlying the findings in their manuscript fully available?

Reviewer #1: Yes

Reviewer #2: Yes

4. Is the manuscript presented in an intelligible fashion and written in standard English?

Reviewer #1: No

Reviewer #2: Yes

5. Review Comments to the Author

Reviewer #1: 1. The authors were recommended to ask professional editors or native speaker to polish the manuscript language due to that many sentences were difficult to follow. E.g., “The data was based on the travel survey data collected from the residents in Xianyang in 2018 and the POI data…”

2. I am not so convinced about the statement “Traffic generation and attraction forecast are the first element of the traditional four stage prediction method, which can be classified into growth rate method, original unit method (cross classification or type analysis) and function method…” Please add more explanations.

3. The authors were recommended to better summarize the academic contributions in the INTRODUCTION section.

4. The authors were recommended to merge the study area section and data collection section.

5. Please delete Chinese symbol 元 in table 1 considering that the journal is an international journal with potential readers were around the world.

6. Pleas number each equation in the manuscript.

7. The following studies were recommended to be properly cited: [1] Traffic flow prediction by an ensemble framework with data denoising and deep learning model, Physica A: Statistical Mechanics and its Applications, vol. 565, p. 125574, 2021.[2] Sensing Data Supported Traffic Flow Prediction via Denoising Schemes and ANN: A Comparison, IEEE Sensors Journal, vol. 20, pp. 14317-14328, 2020.

Reviewer #2: I find the idea has certain application value; Following are my comments after reading your manuscript.

1. It is suggested to further explain the consistency between the groups of the questionnaire and the actual situation of urban citizens.

2. Figure 2-figure 10. These nine figures are just a simple list in the main body of the paper. It is suggested to add corresponding discussion and explanation.

3. This paper constructs three models, but the construction premise of the three models is not clear, and the relationship and research value of the three models are not explained in detail.

4. The innovation and practical application scenarios of the paper need to be further elaborated.

I hope my comments may be useful for the authors.

Best wishes.

6. PLOS authors have the option to publish the peer review history of their article (what does this mean?). If published, this will include your full peer review and any attached files.

Reviewer #1: No

Reviewer #2: No

---

## [Author Response · Author response to Decision Letter 0]

6 Apr 2022

Dear Editors and Reviewers:

Thank you for your letter and for the reviewers’ comments concerning our manuscript entitled “Built Environment Impact on the Per Capita Cycling Frequency of family”(ID: PONE-D-21-40172). Those comments are all valuable and very helpful for revising and improving our paper, as well as the important guiding significance to our researches. We have studied comments carefully and have made correction which we hope meet with approval. The changes in the revised manuscript have been highlighted by red color. Point by point responses to the reviewers’ comments are listed below this letter.

Responds to the reviewer’s comments:

TO REVIEWER 1：

Response:

Thank you very much for the detailed comments. We appreciate your time and help in reviewing our manuscript, and the insightful comments you provided that have helped significantly improve the quality of this study. We have revised the paper very carefully according to your suggestions, and detailed explanations of all the issues are as follows.

1. The authors were recommended to ask professional editors or native speaker to polish the manuscript language due to that many sentences were difficult to follow. E.g., “The data was based on the travel survey data collected from the residents in Xianyang in 2018 and the POI data…”

Response:

Many thanks for pointing out this issue. 

We apologize for the poor language of our manuscript. We have now worked on both language and readability and we also leverage 3rd party service for language polishing. The English of the full manuscript has been strongly revised in the following aspects:

1. We have improved the language expression logic to guide the readers better.

2. We have reorganized language expressions for the verbose passages to make it easier for readers to understand.

3. We have revised the grammar and typos in the original text.

4. We have enriched vocabulary expressions.

5. We have tried to avoid the use of first-person pronouns.

All modified parts are marked in red font in the updated manuscript. 

2. I am not so convinced about the statement “Traffic generation and attraction forecast are the first element of the traditional four stage prediction method, which can be classified into growth rate method, original unit method (cross classification or type analysis) and function method…” Please add more explanations.

Response:

Many thanks for your question. We elaborated this issue as follows:

I am sorry that this part was not clear in the original manuscript. We have reorganized this passage with reference to relevant literature as follows:

Trip generation forecast is the first stage of the four-step traffic demand forecast method. There are many approaches for trip generation forecast, of which the most commonly used ones are category generation model, growth rate method, regression generation model, etc.[1,2] However, the above model often relies on the aggregate prediction method to forecast trip generation and attraction in each traffic zone. Therefore, it is difficult to reflect the impact of the socioeconomic attributes of individuals and families on the travel frequency[3].

[1] Liu An, Song Weifang, Yang Peikun. Reaearch on travel generation prediction model. Journal of Tongji University(Natural Science). 1998.26(3):290-293.

[2] Shi Fei, Wang Wei, Lu Jian. Induction and innovation of resident trip generation prediction methods. 2005,3(1):43-46.

[3] Yang Min, Chen Xuewu, Wang Wei et al. Commuters trip generation model based on activity patterns. Journal of Southeast University( Natural Science Edition) 2008, 38 ( 3) :525-530.

The modified contents have been marked in red font on Page 3 line 48-51 and Page 4 line 69-71 in the updated manuscript.

3. The authors were recommended to better summarize the academic contributions in the INTRODUCTION section.

Response:

Thanks for your suggestion.

We have rewritten the Introduction as suggested by the reviewer. According to the different basic travel units of research, we have discussed the aggregate model constructed by traffic zones and the disaggregated model constructed by individuals or households, separately. At the same time, we have added relevant research on machine learning model, with new references of [3,4,10,20,21]. In addition, we have reorganized the academic contributions of this research from the aspects of research perspectives, research elements and research methods.

At research perspectives aspect, this research focuses on the travel generation prediction of cyclists instead of all-mode resident travel generation prediction model. It is committed to studying the correlation mechanism between the per capita cycling frequency of household, household attributes and built environment elements, which promotes green and healthy lifestyle.

As far as research elements are concerned, the built environment elements of existing research mainly include population density, traffic facility density, POI diversity and other conventional indicators. This research has not only included two variables of motor and non-motor vehicle separate railing ratio and non-motor lane parking ratio that affect cycling safety, but also included some factors that are closely related to daily life and have a significant impact on the per capita cycling frequency of household, such as the density of education, shopping, leisure and medical facilities.

From the aspect of research methods, this research has broken through the defect of losing data information in the disaggregating or aggregating process of traditional regression models. According to the nested structure of family-community data and considering the within-group homogeneity and between-group heterogeneity of the data, a multi-level linear regression model was used in this research to explore the effects of household and community level variables on the per capita cycling frequency of household and the cross-layer interaction effects between household and community.

The modified contents have been marked in red font on page 5 line 110-112 and page 6 line 113-119 in the updated manuscript.

4. The authors were recommended to merge the study area section and data collection section.

Response:

Many thanks for your suggestion. 

We have merged the area section and data collection section according to the comments of the reviewer, and the chapter number was readjusted accordingly.

The modified contents have been marked in red font on Page 7 line 135-154 and Page 8 line 155-160 in the updated manuscript.

5. Please delete Chinese symbol 元 in table 1 considering that the journal is an international journal with potential readers were around the world.

Response:

Many thanks for pointing out this issue. 

We apologize for the typos of our original manuscript, in the revised manuscript, we have corrected the figure and revised “元” to CNY.

The modified contents have been marked in red font on Page 8 line 169 and Page 6 line 173-175 in the updated manuscript.

6. Pleas number each equation in the manuscript.

Special thanks to you for your suggestion.

We have numbered each equation in the updated manuscript.

The modified contents have been marked in red font on Page 11 line 223-225, Page 14 line 291-292 and line 308, Page 15 line 309-318 and line 330, Page 16 line 333,347-352 and Page 17 line 352-358 in the updated manuscript.

7. The following studies were recommended to be properly cited: [1] Traffic flow prediction by an ensemble framework with data denoising and deep learning model, Physica A: Statistical Mechanics and its Applications, vol. 565, p. 125574, 2021.[2] Sensing Data Supported Traffic Flow Prediction via Denoising Schemes and ANN: A Comparison, IEEE Sensors Journal, vol. 20, pp. 14317-14328, 2020.

Special thanks to you for your good suggestion. 

Reference:

The paper you provide has a great effect on a comprehensive understanding of the method of forecasting trip generation, so we added this article as reference [20,21] in the revised manuscript, and We will explore this aspect in the next work.

The modified contents have been marked in red font on page 5 line 99-107 in the updated manuscript.

TO REVIEWER 2:

Response:

We appreciate your time and help in reviewing our manuscript. Thank you very much for your affirmation of our work in this paper and many thanks for your detailed comments. We have revised the paper very carefully according to your suggestion, and all the modified contents have been marked in red font in the updated manuscript.

1. It is suggested to further explain the consistency between the groups of the questionnaire and the actual situation of urban citizens.

Response:

Many thanks for pointing out this issue. 

This study based on the data from 2018 Daily Trip Survey in Xianyang, China. The total sample size is 7,070 persons of 2,934 households with an average population of 2.41, including 48.8% males and 51.2% females. As for the age structure, the main group is 15–59 years old(81.3%), followed by the age over 60 years old(11.5%) and 0–14 years old(7.2%).

At present, the built up area of Xianyang city is mainly in Qindu District. According to the main data of the seventh national census of Qindu District (the well-developed city center of Xianyang) [1], the average population of each family is 2.42, including 49.47% males and 50.53% females. As for the age structure, the main group is 15–59 years old(66.61%), followed by the age over 60 old(17.89%) and 0–14 years old(15.5%).

Comparing the above two groups of data, it can be obtained that the survey sample is basically consistent with the Seventh Census Data in average household size and gender ratio. Due to the impact of safety and physical strength on cyclists, people aged between 18 and 45 are the main force of cycling[2]. The age distribution of the sample is consistent with that of the actual bicycle users. Therefore, our sample is consistent with the actual situation of urban citizens.

[1]The main data of the seventh national census in Qindu District http://www.snqindu.gov.cn/html/zwgk/xxgkml/zdlyxxghk/tjxx/202106/47607.html

[2] iimedia Research. 2018 Special research on the development status of Shared bikes in China. https://www.iimedia.cn/c400/63243.html.2018

2. Figure 2-figure 10. These nine figures are just a simple list in the main body of the paper. It is suggested to add corresponding discussion and explanation.

Response:

Thanks for your valuable suggestion. 

We have added the discussion and explanation of Figure 2-figure 10 in the revised manuscript according to your suggestion.

As can be seen from S2, the population density gradually decreases from the center to the periphery, and the population density in the areas with dense commercial service facilities is higher than that in the new urban areas. It can be seen from S3 and S4 that both the POI diversity and POI density show a typical pattern of contiguous distribution and spread along the line. Contiguous distribution mainly includes the old street area, the business circles on Maotiao Road, the business circles of 505 Square an others, while the spread line mainly includes Renmin Road, Yuquan Road, Weiyang Road, etc. As for the new urban areas, both the diversity and density of POI are low.

Seen from S5, S6, S7 and S8, the densities of road network, intersection, bus station and public service facility around the old urban areas and commercial core areas are higher than those of the urban fringe areas and newly built areas. S8 and S9 show that the traffic volume of the commercial core areas is large and most of the roads are equipped with motor and non-motor vehicle separate railings. The roads of the new areas in the west and south were built under higher construction standards with cross section of three or four boards roads in the main and secondary roads, therefore, the motor and non-motor vehicle separate railings account for a relatively high proportion. From S10, it can be seen that the proportion of non-motor lane parking in the western part of the built-up area is more serious than that in the eastern part. For one thing, there are relatively few traffic monitoring facilities in the west and the law enforcement is weaker than that in the east. For another, there are fewer public parking lots in the west areas compared with the east areas.

The modified contents have been marked in red font on Page 11 line 237-241, Page 12 line 242-256 in the updated manuscript.

3. This paper constructs three models, but the construction premise of the three models is not clear, and the relationship and research value of the three models are not explained in detail.

Response:

Many thanks for pointing out this issue. 

We have supplemented and improved the construction premises, interrelation and research values of these three models in detail.

1、The construction premises of the three models

Traditional linear models, such as variance analysis model and regression analysis model, can only analyze problems involving one layer of data, but cannot comprehensively analyze problems involving nested and hierarchical data. In many studies, the sampling often comes from different levels and units, which brings many cross-level (multi-layer) research problems, thus the multi-layer linear models can be used to effectively and statistically to solve these problems.

A random effect variable can be added to the multi-layer model to make the multi-layer model work well for internal correlation data and data with unequal variance. The multi-layer model assumes that the level 1 residual follows a normal distribution and the level 2 residual follows a multivariate normal distribution, and that the level 1 and level 2 residuals are independent of each other.

2、the relationships of the three models 

At the beginning of the study, no one knows for sure which model could fit the data most satisfactorily, could provide meaningful and interpretable results and could be a parsimonious model at the same time. Generally speaking, modeling is usually an exploratory process based on both statistical and theoretical considerations. Similarly, modeling a multi-layer model is also a step-by-step process. Combining the methods recommended by Hox (1994) and Singer (1998)[3,4], this research firstly fit an empty model to examine the outcome variability. Then, it sequentially examined the macro and micro explanatory variables that may affect the outcome variability and their cross-level interactions.

The empty model is the basis of modeling a multi-level model, because it can provide an estimate of the correlation coefficient within the data group so as to judge whether it is necessary to model a multi-level model. It is only necessary to continue the modeling of a multi-layer model after determining that there is a significant correlation within the data group. Therefore, it is always the first step to run the empty model in the process of modeling a multi-layer model.

The next step in modeling a multi-layer model is to add level-1 (Hox, 1994) or level-2(singer, 1998) explanatory variables into the empty model. We prefer the approach of Singer (1998) to extend the model by adding level-2 explanatory variables. If the results of the empty model show that the data has significant within-group correlation or within-group homogeneity, it means that the data has between-group heterogeneity. Thus, the between-group variation in the mean outcome remains to be explained. Logically speaking, the next step in modeling is to include group-level variables in the model to explain this variability.

The individual characteristic, that is, the level-1 explanatory variable was not controlled when examining the relationship between the group outcome means and the group-level variables. In the third step of modeling, we added level-1 variables to the model and treated all the slopes of level-1 variables as fixed. The set models in the modeling steps 2 and 3 have the same random effects, but different fixed effects.

After the third step of the modeling exploration, we determined which slopes of the level-1 variables in the model are random coefficients in order to test the complete model with both level-1 and level-2 variables. The complete model can account for or explain how the overall outcome variables are affected by the first and second layer factors with theoretical construction. Also, the slopes of level-1 variables can used as a function of the group level explanatory variables in the macro model for cross-layer interaction to explain the between-group variation at a group level.

2、Research value

First of all, the traditional linear model can only analyze the problems involving a certain level of data, rather than comprehensively analyze the problems involving two or more levels. The multi-level linear model is an effective statistical method to solve these problems. This model can not only study the effect of individual level and group level variables on outcome measurement, but also provide an opportunity to study the variation of outcome measurement at different levels.

Secondly, the multi-layer linear model can deal with missing values on the basis of maximum likelihood or restricted maximum likelihood estimation. Therefore, the original data do not need to remove the research objects with missing values, nor need to make up for the missing observation values.

Finally, the multi-layer linear model does not require the observation values to be independent of each other, so it can correct the parameter standard error estimation bias caused by the non-independent observation data. The parameter estimation method of the model is more stable and the prediction results are more effective.

[3]Hox, J. J. 1994. Applied Multilevel Analysis. Amsterdam. TT-Publikaties.

[4]Singer, J. D. 1998. Using SAS Pro Mixed to fit multilevel models, hierarchical models, and individual growth models. J. of Educ. Behav. Stat. Vol.24: 323-255.

The modified contents have been marked in red font on Page 13 line 279-288 in the updated manuscript.

4. The innovation and practical application scenarios of the paper need to be further elaborated.

Response:

Many thanks for pointing out this issue. 

The innovation of this research was added at the beginning of the sixth paragraph and How the study is useful for practical purpose was added at the end of Conclusion as one paragraph.

At research perspectives aspect, this research focuses on the travel generation prediction of cyclists instead of all-mode resident travel generation prediction model. It is committed to studying the correlation mechanism between the per capita cycling frequency of household, household attributes and built environment elements, which promotes green and healthy lifestyle.

As far as research elements are concerned, the built environment elements of existing research mainly include population density, traffic facility density, POI diversity and other conventional indicators. This research has not only included two variables of motor and non-motor vehicle separate railing ratio and non-motor lane parking ratio that affect cycling safety, but also included some factors that are closely related to daily life and have a significant impact on the per capita cycling frequency of household, such as the density of education, shopping, leisure and medical facilities.

From the aspect of research methods, this research has broken through the defect of losing data information in the disaggregating or aggregating process of traditional regression models. According to the nested structure of family-community data and considering the within-group homogeneity and between-group heterogeneity of the data, a multi-level linear regression model was used in this research to explore the effects of household and community level variables on the per capita cycling frequency of household and the cross-layer interaction effects between household and community.

To sum up, diversity of public service facilities, good road infrastructure and appropriate policies are conductive to the cycling safety and comfort for community members to promote cycling. Same specific measures are as follows: firstly, on the urban construction, reasonable arrangement of education, medical and other facilities adjacent to home areas can ensure that daily travel distances of residents are within a proper range of cycling. Secondly, it is necessary to improve the safety of cycling environment by establishing bicycle lanes and other facilities. Finally, if the existing built environment is difficult to optimize, policies such as increasing the barrier to purchasing cars shall be considered, so as to curb the use of motor vehicles and promote active travel behavior. 

The modified contents have been marked in red font on Page 5 line 110-112, Page 6 line 113-119 and Page 27 line 552-560 in the updated manuscript.

---

## [Decision Letter · Decision Letter 1]

19 Apr 2022

Built Environment Impact on the Per Capita Cycling Frequency of Family

PONE-D-21-40172R1

Dear Dr. Zhang,

We’re pleased to inform you that your manuscript has been judged scientifically suitable for publication and will be formally accepted for publication once it meets all outstanding technical requirements.

Kind regards,

Quan Yuan, Ph.D.

Academic Editor

PLOS ONE

Additional Editor Comments (optional):

Reviewers' comments:

Reviewer's Responses to Questions

**Comments to the Author**

1. If the authors have adequately addressed your comments raised in a previous round of review and you feel that this manuscript is now acceptable for publication, you may indicate that here to bypass the “Comments to the Author” section, enter your conflict of interest statement in the “Confidential to Editor” section, and submit your "Accept" recommendation.

Reviewer #1: All comments have been addressed

Reviewer #2: All comments have been addressed

2. Is the manuscript technically sound, and do the data support the conclusions?

Reviewer #1: Partly

Reviewer #2: Yes

3. Has the statistical analysis been performed appropriately and rigorously? 

Reviewer #1: Yes

Reviewer #2: Yes

4. Have the authors made all data underlying the findings in their manuscript fully available?

Reviewer #1: Yes

Reviewer #2: Yes

5. Is the manuscript presented in an intelligible fashion and written in standard English?

Reviewer #1: Yes

Reviewer #2: Yes

6. Review Comments to the Author

Reviewer #1: My comments have been welll addressed, and thus I recommend the editor to accept the wondoerful manuscript.

Reviewer #2: The author has carefully revised the paper according to the review opinions. I recognize the research work and innovation of the paper, and suggest that it can be published in this journal.

7. PLOS authors have the option to publish the peer review history of their article (what does this mean?). If published, this will include your full peer review and any attached files.

Reviewer #1: No

Reviewer #2: No

---

## [Editor Report · Acceptance letter]

4 May 2022

PONE-D-21-40172R1 

Built Environment Impact on the Per Capita Cycling Frequency of Family
——Based on Two-level Hierarchical Linear Model 

Dear Dr. Zhang:

I'm pleased to inform you that your manuscript has been deemed suitable for publication in PLOS ONE. Congratulations! Your manuscript is now with our production department. 

Kind regards, 

on behalf of

Dr. Quan Yuan 

Academic Editor

PLOS ONE